# Reinforcement Learning for Symbolic Graphics Code Generation with Visual Feedback

## Abstract

Symbolic graphics code generation, particularly text-to-SVG generation, plays a critical role in numerous practical applications, including web design, digital publishing, and user interface prototyping. However, current open large language models face significant challenges in handling these visually intricate and structurally precise tasks, often exhibiting a considerable performance gap compared to leading proprietary models. In this paper, we present a novel approach aimed at substantially improving the capabilities in text-to-SVG tasks. Our main contributions are threefold: First, we propose a reinforcement learning framework that leverages vision-language models (VLMs) as visual reward model, providing comprehensive visual feedback that guides LLMs towards generating more accurate and visually coherent SVG outputs. Second, we investigate inference-time scaling methods through extended long Chain-of-Thought (CoT) reasoning combined with large-scale RL, revealing that such methods inherently counteract reward hacking by refining prompt engineering and making task objectives more explicit and concrete. Third, we introduce a new, high-quality benchmark alongside a rigorously curated training dataset dedicated to text-to-SVG generation, addressing the notable absence of specialized benchmarks and datasets in this domain. Experiments on open model, i.e., Qwen3 demonstrate that our approach achieves results comparable to state-of-the-art proprietary and larger models, including Claude-4.0-Sonnet. This work substantially narrows the performance gap and provides both methods and resources to advance symbolic code generation research.

## 1 Introduction

Large language models (LLMs), have made remarkable progress across a wide range of domains (Hurst et al., 2024; Jaech et al., 2024; OpenAI, 2025c; Team et al., 2023), including question answer (Ouyang et al., 2022; Yang et al., 2025a; Team et al., 2024), code generation (Hui et al., 2024), and complex problem-solving (Guo et al., 2025; Wang et al., 2024b; Yang et al., 2024; Team, 2025). Nevertheless, generating symbolic graphics code, particularly from natural language to Scalable Vector Graphics (SVG), remains a persistent challenge (Nishina & Matsui, 2024). Unlike conventional code generation, text-to-SVG tasks require not only syntactic correctness but also adherence to structural precision and visual semantics (Cai et al., 2023). Bridging the gap between textual descriptions and visually faithful SVG would significantly advance a variety of practical applications, including digital publishing, web design, educational illustration, and user interface prototyping.

Although proprietary models have achieved strong performance in symbolic graphics tasks (Yang et al., 2025b), open-source models still fall noticeably behind. One major factor underlying to this gap is the lack of visual feedback during pre-training and post-training. Many current methods focus primarily on textual correctness, overlooking the visual quality of the rendered outputs. Yet in symbolic graphics generation, success is ultimately determined by how well the generated image reflects the intended meaning of the input. Without explicit visual feedback, models often produce outputs that are syntactically correct but visually misaligned with the user's instructions.

To address this limitation, we propose a reinforcement learning (RL) framework that incorporates visual feedback into the post-training of LLMs. Central to our approach is the use of frozen vision-language models (VLMs) (Bai et al., 2025; OpenAI, 2023; Lu et al., 2024; Wu et al., 2024b) as perceptual judges, which evaluate the alignment between the rendered SVG output and the reference

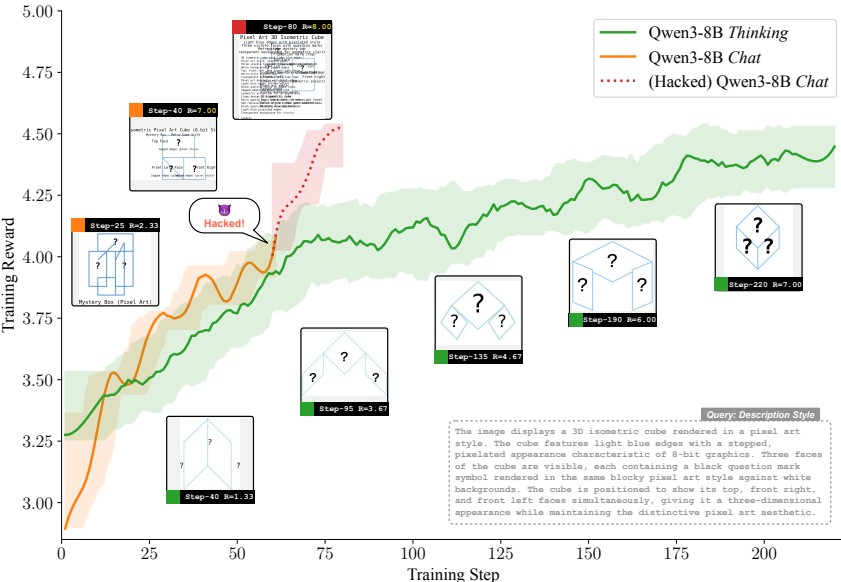

Figure 1: Training curves for Qwen3-8B under different settings. The *Chat* model (orange) initially improves but exhibits reward hacking (red dotted) by inserting descriptive text instead of graphical elements. In contrast, the *Thinking* model (green), enabled by inference-time reasoning, maintains consistent gains without reward hacking, producing visually faithful SVG code.

image. Rather than relying on token-level or syntactical correctness, the reward signal is grounded in perceptual fidelity across multiple visual dimensions (Baumli et al., 2023). These include object presence and accuracy, spatial arrangement, and overall stylistic quality (Rodriguez et al., 2023). The resulting reward functions are not only sensitive to visual details but are also fine-grained, enabling scalable optimization. Compared to conventional methods, our approach introduces a stronger inductive bias toward generating SVG code that is both semantically faithful and visually coherent.

However, the introduction of perceptual rewards also brings new challenges. As shown in Figure 1, reward hacking is observed (Weng, 2024), where the model exploits weaknesses in the reward signal, such as embedding descriptive text into the image in place of rendering the graphical elements. This behavior is largely driven by biases in vision-language models pretraining stage, many of which are pretrained with grounding data that assigns comparable importance to textual overlays and visual features (Wang et al., 2024a; Bai et al., 2025). To mitigate such failure modes, we investigate inference-time scaling as a complementary strategy (Team et al., 2025; Guo et al., 2025; Jaech et al., 2024; OpenAI, 2025c; Yang et al., 2025a). Specifically, we employ reasoning model and prompt it to engage in long chain-of-thought (CoT) decomposition prior to code generation (Yang et al., 2025a; Guo et al., 2025). This "thinking mode" encourages the model to reflect on the instruction and internalize task constraints before producing code. We find that such reflective generation significantly reduces reward hacking and leads to improved output quality across both visual fidelity and instruction compliance.

In parallel with these technical contributions, we also address a data limitation in this domain. Existing benchmarks are either absent or insufficiently tailored to capture the nuances of symbolic graphics generation. To fill this gap, we construct a high-quality benchmark and a curated training dataset using a scalable pipeline. Our process ensures both structural diversity and semantic alignment between code, rendered images, and textual instructions, supporting reliable evaluation and training.

In summary, we present a unified framework for symbolic graphics code generation that integrates perceptual feedback and inference-time scaling. Applied to open-source models such as Qwen3 (Yang et al., 2025a), we substantially narrows the performance gap with state-of-the-art proprietary models. Beyond demonstrating empirical gains, this work contributes scalable methodologies and resources that advance the broader goal of aligning language models with visually grounded semantics.

## 2 TASK FORMULATION

Let $\mathcal{Q}$ be the space of natural language query (instructions), and let $\mathcal{O}$ denote the set of syntactically valid SVG code. The text-to-SVG task is to learn a mapping $\pi_\theta : \mathcal{Q} \to \mathcal{O}$, parameterized by $\theta$, that generates an SVG program $o \in \mathcal{O}$ given an input instruction $q \in \mathcal{Q}$. Each SVG code $o$ is rendered into an image $I \in \mathcal{I}$ via a deterministic function $\mathrm{render} : \mathcal{O} \to \mathcal{I}$. The goal is to learn a policy $\pi_\theta$ that produces code $o$ whose rendered image $I = \mathrm{render}(o)$ faithfully reflects the visual semantics described by $q$. Due to the inherent ambiguity of the task, multiple distinct codes may yield visually identical or semantically equivalent images. As a result, evaluation is conducted primarily in the visual domain, rather than based on token-level similarity between codes. This task is motivated by real-world applications such as web design, UI/UX, and digital publishing, where SVGs provide a compact and resolution-independent representation of icons, charts, and other vector graphics.

## 3 METHOD

### 3.1 PRELIMINARY

**Group Relative Policy Optimization (GRPO)** Reinforcement learning excels at code and mathematical reasoning tasks (Guo et al., 2025; Jaech et al., 2024; Yang et al., 2025a), where programmatic oracles (e.g., unit tests, symbolic solvers) provide explicit correctness signals. Text-to-SVG generation lacks such ground truth: many syntactically different codes render the same image, and textual metrics correlate poorly with visual fidelity. Unlike actor–critic methods such as PPO (Schulman et al., 2017) that learn a separate critic, GRPO (Shao et al., 2024) eliminates the need for a critic by comparing a group of completions directly. For a query $q$, the behavior policy $\pi_{\theta_{old}}$ samples a group of $G$ completions $\{o_i\}_{i=1}^G$. Each completion $o_i$ receives a task-specific reward $R_i = R(q, o_i)$. Its group-relative advantage is then the computed via z-score normalization:

$$\hat{A}_i \;=\; \frac{R_i - \mathrm{mean}\big(\{R_j\}_{j=1}^G\big)}{\mathrm{std}\big(\{R_j\}_{j=1}^G\big)}. \tag{1}$$

Policy improvement is performed at the token level. Let $o_{i,t}$ denote the $t$-th token of code $o_i$. GRPO maximizes a clipped surrogate objective with a KL regularization term:

$$J_{\mathrm{GRPO}}(\theta) = \mathbb{E}_{(q,a)\sim\mathcal{D}}\Big[\mathbb{E}_{\{o_i\}_{i=1}^G \sim \pi_{\theta_{old}}(\cdot|q)}\Big[\frac{1}{G}\sum_{i=1}^G \frac{1}{|o_i|}\sum_{t=1}^{|o_i|}\min\Big(r_{i,t}(\theta)\,\hat{A}_i,\; \mathrm{clip}\big(r_{i,t}(\theta), 1-\varepsilon, 1+\varepsilon\big)\,\hat{A}_i\Big)\Big]$$
$$-\,\beta\,D_{\mathrm{KL}}\big(\pi_\theta \parallel \pi_{\mathrm{ref}}\big)\Big], \tag{2}$$

where the importance ratio $r_{i,t}(\theta)$ is defined as:

$$r_{i,t}(\theta) = \frac{\pi_\theta(o_{i,t} \mid q, o_{i,<t})}{\pi_{\theta_{old}}(o_{i,t} \mid q, o_{i,<t})}. \tag{3}$$

Here, $\varepsilon$ is the clipping threshold, $\beta$ controls the strength of the KL regularization toward a fixed reference policy $\pi_{\mathrm{ref}}$, and expectations are taken first over tokens, then averaged over the $G$ sampled codes. GRPO thus optimizes the policy toward completions that outperform others.

### 3.2 REINFORCEMENT LEARNING WITH VISUAL FEEDBACK

Reinforcement learning excels in code and math reasoning because deterministic oracles supply explicit correctness signals (Guo et al., 2025). However, text-to-SVG generation lacks such ground-truth supervision. A visually accurate SVG may have multiple structurally distinct implementations, and textual similarity metrics often fail to reflect perceived visual fidelity. We overcome this gap with a frozen state-of-the-art vision-language model (VLM). At each training step, the model generates SVG code, we render it, pair the image with the prompt, and query VLM for a score. This dense scalar reward proxies human preference, pushing the model toward syntactically valid, semantically faithful, and visually coherent graphics.

---

**Prompt for Visual-Langauge Model**

Please help me evaluate SVG images against specified instructions and a reference image through three major assessment areas. Each area is scored independently, with scores summed for a final rating.
 - **First Image**: the generated image to be evaluated
 - **Second Image**: the reference "ground-truth" image
Here's the specified instructions for SVG code writing:
{THE_SVG_INSTRUCTION}
**1. Object and Text Accuracy (0-3 points)**
**Criteria**: Object Presence, Object Completeness, Shape Accuracy, Text Accuracy, Typography
 - **0 points**: Significant deviation from requirements, critical objects missing or severely distorted
 - **1 points**: [Base GOOD score] All required objects present and identifiable, though may have minor flaws in shape or execution
 - **2 points**: [Perfect shapes] All objects match specified shapes, with correct proportions and proper sizing relative to each other
 - **3 points**: [Outstanding accuracy] Perfect shaping with precise edges, and perfect text implementation matching reference exactly
**2. Positioning and Stroke Precision (0-4 points)**
**Criteria**: Relative Positioning, Size and Proportion, Stroke Accuracy, Clean Layout, Viewbox Utilization
 - **0 points**: Completely incorrect layout or missing stroke elements
 - **1 points**: Significant layout issues, problematic stroke implementations or poor positioning with major overlapping issues
 - **2 points**: [Base GOOD score] Objects positioned correctly with proper spacing and appropriate stroke weights
 - **3 points**: [Excellent positioning] Perfect layout matching reference image with precise spacing and optimal viewbox utilization
 - **4 points**: [Masterful execution] Perfect positioning that match reference image with exceptional accuracy down to the pixel level
**3. Color and Overall Quality (0-3 points)**
**Criteria**: Color Matching, Opacity/Transparency, Rendering Quality, Detail Precision, Overall Impression
 - **0 points**: Incorrect colors, severe rendering failures or major quality problems
 - **1 points**: [Base GOOD score] Colors match specifications, rendering is clean with no artifacts
 - **2 points**: [Perfect coloring] Exact color matching to reference image, with appropriate use of opacity/transparency if specified
 - **3 points**: [Outstanding quality] Perfect color implementation that perfectly matches or exceeds reference image
**Evaluation Guideline**
 - Reference the second image (the reference "ground-truth" image) when assessing
 - Always compare to the reference when assigning higher scores than the [Base GOOD score]
 - If an element does not match the reference image, it must be noted and reflected in the score
*Please write your evaluation in the following format:*
```xml
<comparison_summary>
...brief overall comparison between the generated image and the reference image...
</comparison_summary>
<object_text_accuracy> <review>...</review> <score>...a integer...</score>
</object_text_accuracy>
<positioning_stroke> <review>...</review> <score>...</score> </positioning_stroke>
<color_overall> <review>...</review> <score>...</score> </color_overall>
<final_score>...a integer...</final_score>
```

Figure 2: Prompt used for the VLM judge, specifying evaluation criteria and XML output format.

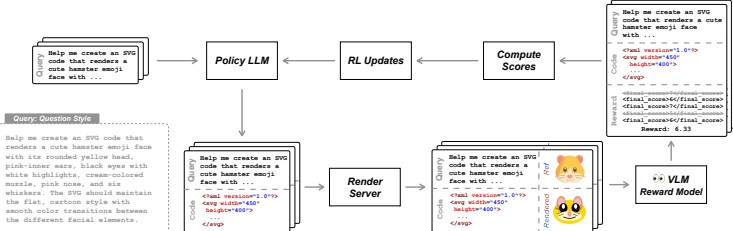

Figure 3: Overview of reinforcement learning with visual feedback for text-to-SVG. The policy model generates SVG based on a textual query, which is then rendered into an image and evaluated by a vision-language model (VLM). The reward provides visual feedback to guide policy updates.

**VLM-as-a-Judge** The evaluation prompt used for reinforcement learning is designed based on three key principles: **(1) Comprehensive coverage.** We decompose "visual quality" into three independent dimensions critical to SVG graphics: (i) *object and text correctness*, (ii) *geometric alignment and stroke accuracy*, and (iii) *color fidelity and overall polish*. Each dimension is supported by a checklist, ensuring that every visible flaw maps to a concrete criterion. **(2) Fine-grained scoring.** Each dimension is rated on a short integer scale with human-interpretable levels (e.g., adequate, perfect, excellent). The overall score ranges from 0 to 10, with each increment representing a meaningful visual improvement. This dense feedback provides smooth reward gradients and supports stable policy updates. **(3) Machine-readable output.** The judge returns its decision in a fixed XML format that contains both the written assessments and the numeric scores. The consistent output structure allows direct extraction of rewards during training, eliminating the need for custom parsing and ensuring deterministic behavior across runs.

---

**Mitigating Reward Hacking via Inference-Time Scaling**

**Instruction:**
Help me create an SVG code that renders a simple house with an orange trapezoid roof, white walls, two blue windows, a set of four steps on the left side with purple accents, a green base, and some dashed curved lines on the right side to represent wind or rain.

**Response:**
```
<think>
```
Okay, I need to create an SVG of a simple house based on the instruction. Let's start by breaking down the components. First, the house has **an orange trapezoid roof**. A trapezoid has two parallel sides, so **I'll need to define a polygon or use a path** ... So the walls are white. Let's plan the structure. Let's say the house has a square base, **like a rectangle for the walls** ... So for the roof: trapezoid. **Let's make it large enough** ...
```
</think>
```xml
<svg viewBox="0 0 300 300" xmlns="http://www.w3.org/2000/svg">
  <!-- Orange trapezoid roof -->
  <polygon points="80,100 220,100 250,150 50,150" fill="#FF8C00"/>
  <!-- White walls -->
  <rect x="50" y="150" width="200" height="150" fill="#FFFFFF"/>
  ...
</svg>
```

---

Figure 5: An example of inference-time scaling using Qwen3 with a `<think>` directive, where the model reflects step by step before generating SVG.

These designs yield a reward function that is *comprehensive*, *modifiable*, and *reproducible*, which are essential properties for reliable reinforcement learning in text-to-SVG generation.

**Reward Score**   Each training sample consists of a textual instruction $q$ and its reference SVG code $o^*$. The policy $\pi_\theta$ generates a candidate $o$. Both are rendered into images:

$$I^* = \text{render}(o^*), \qquad I = \text{render}(o). \tag{4}$$

The tuple $(q, I, I^*)$, together with the prompt (Figure 2), is passed to a frozen vision–language model, which returns an integer score $r \in [0, 10]$, labelled as `<final_score></final_score>` in the XML output. To reduce the effect of randomness in VLM outputs, we query the VLM five times ($\mathcal{R} = \{r_1, ..., r_5\}$), drop the highest and lowest scores, and average the rest to define the reward:

$$R_{\text{vlm}}(q, o) = \text{mean}(\mathcal{R} \setminus \{\min \mathcal{R}, \max \mathcal{R}\}). \tag{5}$$

### 3.3  MITIGATING REWARD HACKING VIA INFERENCE-TIME SCALING

**Reward Hacking**   While reinforcement learning based on visual feedback can improve text-to-SVG performance in the early stages of training, we observe that it often leads to reward hacking in later stages (Weng, 2024; Di Langosco et al., 2022), which results in degradation of output quality (Pan et al., 2022; Skalse et al., 2022). A prominent failure mode occurs when the model generates SVGs that include textual annotations describing visual elements, such as "Red Roof" or "Blue Body of the House", instead of rendering the corresponding graphics. These outputs frequently receive high reward scores from the vision-language model, even though they fail to capture the intended visual semantics.

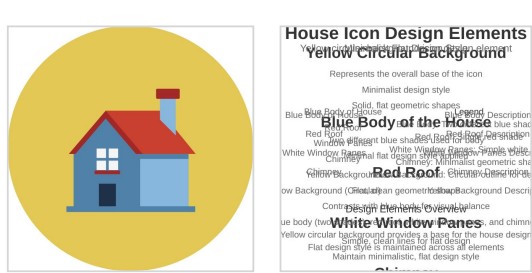

Figure 4: A case of reward hacking where the model embeds descriptive text like "Red Roof" instead of rendering the intended visual concept.

This behavior stems from biases in the reward signal introduced by vision-language models, many of which are pretrained with grounding-based supervision (Wang et al., 2024a; Bai et al., 2025). In such settings, both textual overlays and object-level features contribute similarly during alignment. As a result, rendering relevant words within the SVG becomes a shortcut for maximizing reward, even when the output fails to reflect the intended graphical content.

**Inference-time Scaling**   Accordingly, we choose a thinking-enabled model as the RL policy backbone, using inference-time reasoning to avoid reward hacking and promote visually grounded, semantically coherent outputs. Figure 5 shows our implementation with Qwen3, whose explicit `<think>`

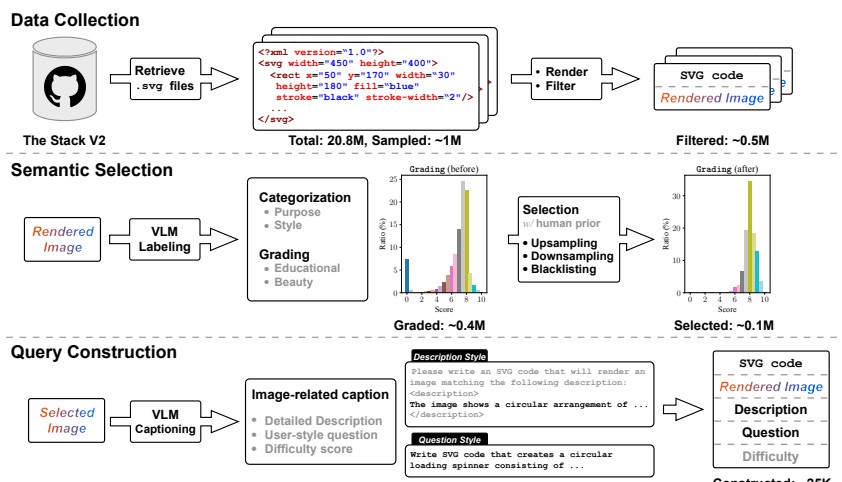

Figure 6: Data pipeline for constructing text-to-SVG training examples.

directive triggers a chain-of-thought (CoT, Wei et al. (2022)) before SVG synthesis. Reflection lets the model parse the instruction, decompose the visual specification, and embed implicit structure and task-specific constraints. For example, it reasons about each SVG component before coding, reducing reliance on reward shortcuts and producing outputs that remain both visually grounded and semantically faithful.

## 3.4 DATA SYNTHESIS PIPELINE

Manual text-to-SVG annotation is prohibitively expensive, demanding design expertise, coding skills, and pixel-level alignment. We instead build a synthetic pipeline that programmatically converts raw SVGs into large, diverse, high-quality instruction–code pairs with minimal human effort (Figure 6). The pipeline has three stages: data collection, semantic selection, and query construction.

**Data Collection.**    We begin by retrieving all SVG files from The Stack V2 (Lozhkov et al., 2024), a permissively licensed code dataset. We randomly sample around 1 million SVG files out of 20.8 million. Files that fail to render, produce blank or corrupt outputs, or have invalid dimensions or extreme aspect ratios are filtered out, resulting in approximately 0.5 million valid code-image pairs.

**Semantic Selection.**    We then apply a frozen vision-language model (VLM) to each rendered image. The model performs two core tasks: (i) semantic categorization, which assigns coarse labels for *purpose* (e.g., UI, chart, icon) and *style* (e.g., flat, line, modern); and (ii) scalar grading, which evaluates each image's educational clarity and visual appeal on a [0–10] scale. From approximately 400K graded samples, we apply a human-informed selection strategy based on labels' combinations. This process yields a curated set of roughly 100K examples with improved quality and diversity.

**Query Construction.**    For each selected image, we generate textual instructions using VLM-based captioning. We synthesize two complementary prompt styles: (i) a **description-style** instruction providing a detailed natural language account of the image, and (ii) a **question-style** instruction emulating typical user queries (e.g., "Create an SVG of..."). A difficulty score is assigned to each sample based on structural and visual complexity. The final dataset contains around 25K instances, each consisting of the SVG code, rendered image, associated textual instructions, and difficulty score, forming tuples of the form ⟨Instruction, SVG Code, Image, Difficulty⟩.

By synthesizing data grounded in both symbolic structure and visual semantics, our pipeline provides scalable, controllable supervision for training LLMs via reinforcement learning in text-to-SVG tasks.

Table 1: Comparison of closed-source and open-source on the text-to-SVG benchmark. Qwen3-8B with proposed RL method achieves competitive performance.

| Model | Thinking | Description | | | | Question | | | |
|---|---|---|---|---|---|---|---|---|---|
| | | Easy | Medium | Hard | Overall | Easy | Medium | Hard | Overall |
| *Closed-Source LLMs* | | | | | | | | | |
| Claude-4-Sonnet | | 5.87 | 5.61 | 4.95 | 5.69 | - | - | - | - |
| Claude-4-Sonnet | ✓ | 5.78 | **5.88** | 4.64 | 5.73 | - | - | - | - |
| Claude-3.7-Sonnet | | **6.08** | 5.80 | 4.21 | **5.81** | 5.67 | 5.23 | 4.08 | 5.36 |
| Claude-3.7-Sonnet | ✓ | 5.89 | 5.71 | 3.59 | 5.64 | **5.76** | **5.48** | 4.28 | **5.53** |
| Claude-3.5-Sonnet | | 5.86 | 5.26 | 3.46 | 5.41 | 5.56 | 4.87 | 2.97 | 5.07 |
| GPT-4.1 | | 5.86 | 5.42 | **5.15** | 5.62 | - | - | - | - |
| GPT-4.5-Preview | | 5.90 | 5.77 | 4.13 | 5.71 | 5.63 | 5.23 | **4.31** | 5.35 |
| ChatGPT-4o | | 5.73 | 5.70 | 4.56 | 5.62 | 5.59 | 5.25 | 3.79 | 5.31 |
| GPT-4o-mini | | 5.02 | 4.79 | 3.87 | 4.83 | 4.73 | 4.01 | 3.46 | 4.33 |
| *Open-Source LLMs* | | | | | | | | | |
| R1-Distill-Llama-70B | ✓ | 4.47 | 3.78 | 2.20 | 4.00 | 4.37 | 3.74 | 2.28 | 3.94 |
| R1-Distill-Qwen-32B | ✓ | 4.35 | 3.85 | 2.23 | 3.97 | 4.33 | 3.44 | 2.02 | 3.77 |
| R1-Distill-Qwen-14B | ✓ | 3.99 | 3.17 | 1.65 | 3.46 | 3.84 | 3.11 | 1.76 | 3.37 |
| R1-Distill-Qwen-7B | ✓ | 1.71 | 1.26 | 0.51 | 1.43 | 1.60 | 1.19 | 0.54 | 1.34 |
| Llama-4-Maverick | | 5.13 | 4.72 | 3.37 | 4.82 | 4.73 | 4.25 | 2.70 | 4.37 |
| Llama-4-Scout | | 4.37 | 3.99 | 2.91 | 4.10 | 4.46 | 3.79 | 2.28 | 4.01 |
| Llama-3.1-70B | | 4.70 | 4.09 | 2.57 | 4.28 | 4.48 | 3.88 | 2.03 | 4.03 |
| Llama-3.1-8B | | 3.31 | 2.64 | 1.58 | 2.89 | 3.20 | 2.62 | 1.09 | 2.79 |
| Qwen2.5-Coder-32B | | 4.82 | 4.43 | 2.78 | 4.49 | 4.66 | 4.07 | 2.44 | 4.24 |
| Qwen2.5-Coder-14B | | 4.43 | 3.73 | 2.38 | 3.97 | 4.33 | 3.63 | 2.03 | 3.85 |
| Qwen2.5-Coder-7B | | 3.98 | 3.24 | 1.75 | 3.50 | 3.68 | 3.10 | 1.61 | 3.27 |
| Qwen3-235B-A22B | | 5.40 | 5.06 | 3.55 | 5.11 | 5.28 | 4.64 | 2.94 | 4.83 |
| Qwen3-235B-A22B | ✓ | 5.28 | 5.18 | 3.52 | 5.10 | 5.18 | 4.70 | 3.34 | 4.83 |
| Qwen3-32B | | 5.13 | 4.69 | 2.67 | 4.75 | 4.91 | 4.36 | 2.66 | 4.50 |
| Qwen3-32B | ✓ | 5.03 | 4.93 | 3.40 | 4.86 | 5.04 | 4.63 | 3.06 | 4.71 |
| Qwen3-30B-A3B | | 4.95 | 4.66 | 2.53 | 4.63 | 4.75 | 4.15 | 2.46 | 4.32 |
| Qwen3-30B-A3B | ✓ | 4.98 | 4.83 | 3.48 | 4.80 | 4.80 | 4.44 | 2.97 | 4.50 |
| Qwen3-14B | | 4.96 | 4.49 | 2.84 | 4.60 | 4.85 | 4.17 | 2.17 | 4.35 |
| Qwen3-14B | ✓ | 4.93 | 4.75 | 3.35 | 4.73 | 4.90 | 4.37 | 2.58 | 4.49 |
| Qwen3-8B | | 4.63 | 4.09 | 2.79 | 4.26 | 4.36 | 3.66 | 2.11 | 3.89 |
| ♣Qwen3-8B | ✓ | 4.78 | 4.51 | 3.12 | 4.54 | 4.80 | 4.15 | 2.34 | 4.33 |
| ♣Qwen3-8B *w/ RL* | ✓ | **5.79** | **5.48** | **4.15** | **5.53** | **5.58** | **5.24** | **3.72** | **5.29** |

# 4 EXPERIMENTS

## 4.1 EVALUATION

**Benchmark**    While emerging datasets like SVG-Bench[1] provide initial resources, the field still lacks a standardized evaluation suite. To address this, we curate a high-quality test set of 300 text-to-SVG instances disjoint from training data, following the filtering pipeline in Section 3.4. After iterative multi-dimensional validation, we retain **164** verified samples. To support detailed performance analysis, we categorize the test set into three difficulty levels: *Easy* (82), *Medium* (69), and *Hard* (13). The difficulty annotations are inherited from the scalar grading stage of our data pipeline, where a vision-language model assigns a complexity score to each image. These automatic scores are further verified. There's two prompt styles for each instance to reflect different forms of user intent: (1) a natural *Description*-style prompt and (2) a *Question*-style prompt.

**Compared Models**    We conducted evaluations using a diverse set of models on our text-to-SVG benchmark. For proprietary closed-source models, we selected the largest variants of the Claude 3.5, 3.7 and 4 Sonnet series (Anthropic, 2023; 2025a;b), with the 3.7 and 4 generation supporting extended reasoning capabilities (referred to as "thinking" models). Additionally, we included OpenAI's GPT-4.1, GPT-4.5-Preview, GPT-4o and GPT-4o-mini (OpenAI, 2025a;b; Hurst et al., 2024) in our assessment. For open-source models, we first evaluated several variants distilled from DeepSeek-R1 (Guo et al., 2025) across different sizes. We also considered Llama 3.1 and the latest Llama 4 series (Grattafiori et al., 2024; AI@Meta, 2024; 2025). Within the Qwen family, we selected the Qwen 2.5 Coder series (Hui et al., 2024) and the Qwen3 series (Yang et al., 2025a) featuring Hybrid Thinking Modes, where the "Hybrid" refers to the ability to control the model's depth of reasoning via prompt-based instructions.

---

[1]Available at: `https://github.com/johnbean393/SVGBench`

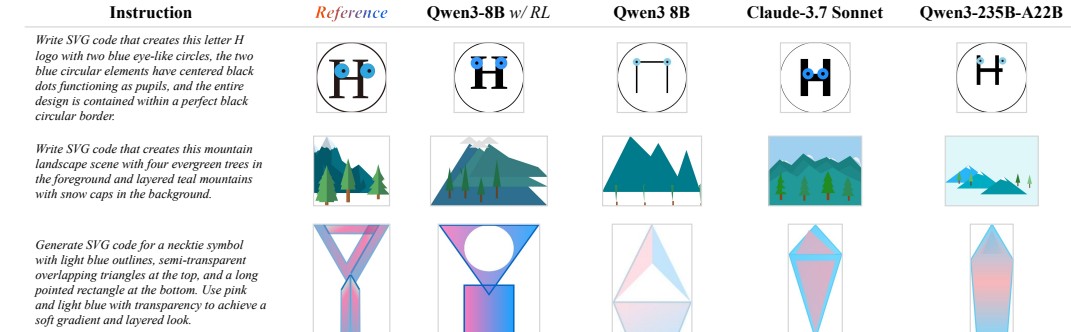

| Instruction | *Reference* | **Qwen3-8B** *w/ RL* | **Qwen3 8B** | **Claude-3.7 Sonnet** | **Qwen3-235B-A22B** |
|---|---|---|---|---|---|
| *Write SVG code that creates this letter H logo with two blue eye-like circles, the two blue circular elements have centered black dots functioning as pupils, and the entire design is contained within a perfect black circular border.* | | | | | |
| *Write SVG code that creates this mountain landscape scene with four evergreen trees in the foreground and layered teal mountains with snow caps in the background.* | | | | | |
| *Generate SVG code for a necktie symbol with light blue outlines, semi-transparent overlapping triangles at the top, and a long pointed rectangle at the bottom. Use pink and light blue with transparency to achieve a soft gradient and layered look.* | | | | | |

Figure 8: Case study comparing text-to-SVG outputs across models.

## 4.2 TRAINING DETAILS

We conduct RL experiments on Qwen3-8B, which supports "Hybrid Thinking Modes" (Yang et al., 2025a), allowing us to investigate the impact of enabling the *Thinking Mode* within the same model. To provide the perceptual reward signal essential for our RL, we employ Qwen2.5-VL 72B (Bai et al., 2025) as the frozen vision-language model judge. For training runs without the *Thinking Mode* (i.e., standard instruction model), we set the maximum generation length to 8,192 tokens. In contrast, for experiments with *Thinking Mode* enabled, the maximum generation length was extended to 16,384 tokens. Each RL training step involved inference over a batch of 256 queries, with 8 rollouts per sample. We set the GRPO mini-batch size to 32 and set clipping parameter to 0.2(Schulman et al., 2017). The learning rate was fixed at 1.5e-6. All other training details followed the standard GRPO algorithm (Shao et al., 2024). All experiments were conducted on 16 NVIDIA A800 80GB GPUs, with a total runtime of approximately 300 hours.

## 4.3 RESULTS

**Main Results** Table 1 shows that closed-source models outperform open-source models on the text-to-SVG benchmark. Within the open-source Qwen family, performance improves with model size, with Qwen3-32B yielding higher scores than 14B and 8B. Moreover, enabling the thinking mode in Qwen models, particularly in smaller ones, brings consistent gains. Notably, incorporating our proposed reinforcement learning method with visual feedback into Qwen3-8B leads to a substantial improvement. The RL-enhanced model achieves an overall score of 5.29, significantly outperforming its supervised counterpart (4.33). This result demonstrates that proposed method can effectively bridge the performance gap, enabling open-source models to approach the capabilities of stronger proprietary models in text-to-SVG tasks. A qualitative comparison in Figure 8 further illustrates the improvement, where RL-trained outputs show greater semantic fidelity and visual completeness.

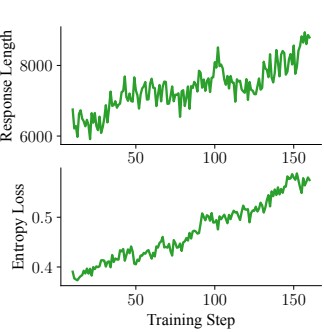

Figure 7: Trends of response length and entropy loss during training with Qwen3-8B in thinking mode.

**Findings** To better understand the behavior of our reinforcement learning method, we monitor two key metrics throughout training: *response length* and *entropy loss*. The results for Qwen3-8B under both *chat* and *thinking* modes are shown in Figure 7. We observe that when trained with proposed method in thinking mode, the model exhibits a steady increase in response length. This suggests that the policy is actively encouraged to perform inference-time scaling during training. At the same time, the entropy loss also increases consistently, indicating a growing degree of exploration. This enables the model to sample potentially higher-reward SVG candidates, further improving learning dynamics.

## 4.4 ABLATION STUDY

As discussed earlier, Qwen3-8B exhibits reward hacking during training. We investigate whether this can be mitigated by **(a)** scaling model size and **(b)** using a code-specialized model. Thus, we conduct training on Qwen2.5-Coder-32B-Instruct, which satisfies both conditions. Figure 9 shows its training reward curve: reward hacking still occurs, though its onset is delayed compared to Qwen3-8B, suggesting that scaling and specialization slow down hacking but do not eliminate it. In our experiments, only enabling the thinking mode consistently prevents reward hacking, highlighting the critical role of inference-time scaling in preserving reward integrity.

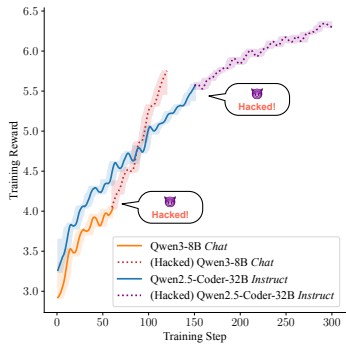

Figure 9: Training reward curves for Qwen3-8B and Qwen2.5-Coder-32B-Instruct.

## 5 RELATED WORK

Recent efforts have explored enhancing LLMs' capabilities in SVG understanding and generation. LLM4SVG (Xing et al., 2024) proposed a modular architecture combining semantic tagging and vector encoders, supported by a 580K-sample dataset. Chat2SVG (Wu et al., 2024a) combined LLMs with diffusion models to generate visually expressive SVGs from text. StarVector (Rodriguez et al., 2023) fused vision encoders and CodeLLMs to generate SVGs. Follow-up study (Rodriguez et al., 2025) have also applied reinforcement learning with rendering feedback, but typically rely on VLM-based image reconstruction and holistic visual similarity for rewards. Several works have also investigated SVG as an evaluation or reasoning medium. SVGEditBench (Nishina & Matsui, 2024) introduced a benchmark for assessing LLMs on structured SVG editing tasks. Cai et al. (2023) demonstrated that LLMs can perform vision-language reasoning via SVG-based representations. However, a common limitation of these reward mechanisms (e.g., FID (Theis et al., 2015), CLIP Score (Radford et al., 2021), FID-CLIP (Wu et al., 2023) or image reconstruction) is their struggle with fine-grained fidelity, as they are ill-suited for a task where a single prompt can have multiple valid visual outputs. In contrast, our method introduces a generative reward model that produces a semantic checklist, enabling fine-grained feedback through RL.

## 6 DISCUSSION

**Limitation and future directions**   This work has several limitations. **First**, due to resource constraints, we only applied our method to Qwen-8B, as larger models require significantly more compute. **Second**, regarding reward hacking, our work explores a different path from directly engineering the reward function. While methods like using perceptual metrics or masking textual artifacts are valid strategies, we focused on improving the model's intrinsic reasoning process. Our findings suggest that the "thinking mode" acts as an effective regularizer, compelling the model to follow instructions more faithfully rather than exploiting reward signals. **Third**, our evaluation relied on a single VLM as the judge. To validate that our improvements are robust and not an artifact of overfitting to a specific judge's biases, we plan to re-evaluate key model outputs using a distinct, powerful VLM (e.g., (InternVL-3 and Qwen3-VL) and report the agreement. **Fourth**, the model still struggles with prompts involving complex spatial reasoning, such as 3D shapes, likely due to insufficient visual feedback. Building on these points, our future work will focus on applying visual feedback–driven reinforcement learning to broader symbolic code domains, such as front-end web generation, while employing a more rigorous multi-judge evaluation framework.

**Conclusion**   In this paper, we introdue a reinforcement learning framework for symbolic graphics code generation, with a focus on text-to-SVG tasks. By leveraging vision-language models as visual reward models, we align model outputs with perceptual semantics. To address reward hacking, we introduce inference-time scaling with thinking-enabled policies, encouraging the generation of visually grounded and semantically faithful code. Furthermore, we construct a high-quality training dataset through a VLM-guided selection and captioning pipeline. These components significantly narrow the performance gap between open-source and proprietary models, and establish a foundation for scalable, perceptually aligned code generation.

## REPRODUCIBILITY STATEMENT

We have taken several steps to support the reproducibility of our work. The construction of our datasets, the design of evaluation prompts, and the overall experimental setup are detailed in Section 3, along with Figures 2, 3, and 6. Section 4 specifies the models used and reports the corresponding evaluation scores. To further facilitate replication, we provide a zipped supplementary material, which include: (1) all prompts used in our experiments, (2) the full evaluation datasets, (3) end-to-end evaluation code described in the paper, and (4) a step-by-step README that guides users to conduct evaluations, which can also be integrated into any standard RL framework.

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

## A  EVALUATION OF GENERAL CAPABILITIES AFTER RL

### A.1  GENERAL CAPABILITIES

A central challenge in model development is the specialization-generalization dilemma: task-specific fine-tuning, while effective for the target domain, often risks degrading a model's broader, general-purpose capabilities. This phenomenon is a significant concern, particularly when using reinforcement learning (RL) on highly structured, synthetic data like text-to-SVG. To investigate this issue, we conducted a comprehensive evaluation of our final model, Qwen3-8B *w/ RL*, against its original base model, Qwen3-8B.

Our evaluation spans a suite of established text-only benchmarks designed to measure foundational abilities, including general knowledge (**MMLU**, Hendrycks et al. 2020), mathematical reasoning (**GSM8K**,Cobbe et al. 2021 and **MATH**Hendrycks et al. 2021), and code generation (**HumanEval**,Chen et al. 2021 and **MBPP**, Austin et al. 2021) with their enhanced version **Hu-manEval+** and **MBPP+** in EvalPlus (Liu et al., 2023). To ensure a fair and thorough comparison, we evaluated both models under two distinct decoding strategies: with and without "thinking mode". The aggregated results are presented in Table 2. We can draw the following conclusions:

Table 2: Performance comparison on general text-based benchmarks. The results demonstrate that RL fine-tuning for SVG generation does not degrade, and in some cases enhances, general capabilities. Higher is better.

| Model | Thinking | MMLU | HumanEval | HumanEval+ | MBPP | MBPP+ | GSM8K | MATH |
|---|---|---|---|---|---|---|---|---|
| Qwen3-8B | ✓ | **89.5** | 89.6 | 81.7 | 85.0 | 69.7 | **94.8** | **48.5** |
| Qwen3-8B *w/ RL* | ✓ | 88.6 | **93.9** | **86.6** | **86.2** | **72.7** | 93.6 | 46.7 |
| Qwen3-8B | | 83.8 | **82.9** | **80.5** | 69.2 | 59.4 | 88.2 | 25.0 |
| Qwen3-8B *w/ RL* | | **84.5** | 82.3 | 79.9 | **71.9** | **60.9** | **91.5** | 25.0 |

**Robust Preservation of Foundational Reasoning and Knowledge**  Across benchmarks, the performance of Qwen3-8B *w/ RL* remains remarkably stable. On challenging tasks like MMLU, GSM8K, and MATH, the scores are either marginally higher or exhibit only negligible decreases (typically within a 1–2 point margin). This stability demonstrates that the specialized training for SVG generation did not impair the model's fundamental language understanding and complex reasoning abilities.

**Clear Evidence of Positive Transfer to Code Generation**  More strikingly, the model after RL shows consistent and significant improvements across multiple code generation benchmarks, including a +4.3 point gain on HumanEval and a +4.9 point gain on HumanEval+ in the "thinking mode". This suggests a powerful positive transfer effect. We hypothesize that the process of learning to generate structured SVG code acts as a beneficial regularizer. This training helps the model learn to follow instructions precisely, think logically, and stick to syntax constraints.

### A.2  TARGETED VALIDATION OF CODING SKILLS

To further probe this observed enhancement in coding capabilities, we performed a more targeted evaluation on **LiveCodeBench** (Jain et al. 2024; versions 5 and 6), a dynamic benchmark for real-world coding challenges. We report **pass@1** and **pass@8** scores in the non-thinking mode.

Table 3: LiveCodeBench results (non-thinking mode), reported as "pass@1 / pass@8". The model after RL maintains or improves performance, corroborating the positive transfer effect on coding tasks.

| Model | LiveCodeBench (v5) | LiveCodeBench (v6) |
|---|---|---|
| Qwen3-8B | **24.2** / 34.1 | 25.9 / **34.3** |
| Qwen3-8B *w/ RL* | 24.0 / **35.3** | **26.8** / 33.5 |

As shown in Table 3, the Qwen3-8B *w/ RL* model maintains highly competitive performance. Notably, it improves the pass@8 rate on v5 and the pass@1 rate on v6, suggesting that the RL training may encourage the model to generate a broader set of correct candidates or improve its first-attempt accuracy, depending on the contest's nature.

In summary, the collective evidence from this multi-faceted evaluation is compelling. Our specialized RL training pipeline for text-to-SVG does not lead to a trade-off. Instead, it enhances the target capability *without sacrificing*—and in the important domain of code generation, even *meaningfully improving*—the model's general-purpose performance. This suggests our approach yields a model that is both a master of its specialized domain and an even more capable generalist.

## B  LLM USAGE STATEMENT

In this work, we used a large language model as a general-purpose writing assistance tool. Specifically, the LLM was used to help refine the clarity, grammar, and overall fluency of the text during the drafting and revision stages. The model provided suggestions for rephrasing sentences, improving paragraph structure, and ensuring consistent academic tone. However, all ideas, research design, analysis and interpretation of results were made by the authors.

