# OpenReview forum: "Reinforcement Learning for Symbolic Graphics Code with Visual Feedback"
_ICLR.cc/2026/Conference — ICLR 2026 Conference Withdrawn Submission_

### Official Review · Reviewer_Qy74 · 2025-10-21

**Soundness:** 3
**Presentation:** 4
**Contribution:** 3
**Rating:** 4
**Confidence:** 5

**Summary:**

The paper tackles text-to-SVG generation with a post‑training reinforcement learning (RL) framework that uses a frozen VLM with carefully constructed prompt as a visual reward model. The authors further show that enabling “Thinking” (long CoT) at inference mitigates VLM‑reward hacking. Extensive experiments are conducted to demonstrate the method’s effectiveness, along with some in-depth analysis.

**Strengths:**

* Clearly articulated motivation for visual evaluation in symbolic graphics code, with practical solutions using structured, machine-readable VLM feedback.
* The finding that Thinking mode reduces hacking is compelling, and the mechanism is plausible: long‑form reasoning clarifies constraints before emitting SVG.
* On the curated benchmark, Qwen3‑8B + RL narrows the gap to much larger models

**Weaknesses:**

* Results rely entirely on a single VLM and prompt set for both evaluation and training, which raises concerns about overfitting and evaluator bias. Incorporating additional metrics would improve reliability.
* The 164‑item test set is produced by the same pipeline used for training data, with subsets with as low as 13, restricting diversity, accuracy and potentially overstating generalization.
* The paper did not compare its vlm based reward method against alternative visual metrics (e.g. image embedding based), limiting insight into the relative effectiveness of their design.
* missing svg specific baselines, e.g. OmniSVG: A Unified Scalable Vector Graphics Generation Model (CVPR 2025)

**Questions:**

During RL, response length increases, it's that increase is from thinking tokens or svg tokens? An extra svg compactness metric is welcomed too.

---

### Official Review · Reviewer_Jojn · 2025-10-28

**Soundness:** 2
**Presentation:** 3
**Contribution:** 2
**Rating:** 4
**Confidence:** 4

**Summary:**

This paper introduces a reinforcement learning method to improve text2SVG generation, aiming to make open-source models competitive with proprietary ones. The approach uses a MLLM as a visual judge, providing detailed perceptual rewards to guide the learning process. The authors make several contributions, including building a new, specialized training dataset and benchmark.

A core finding is their solution to reward hacking. By prompting the model to use chain-of-thought reasoning, or a "thinking mode", they effectively prevent it from cheating by embedding text descriptions instead of rendering graphics. When tested on Qwen3-8B, this method produced results comparable to top models like Claude-4.0-Sonnet, all while preserving the model's general capabilities on other tasks.

**Strengths:**

The paper discovered the problem of training text2SVG directly through visual feedback as a reward, and proposed a method for reward hacking

**Weaknesses:**

1. Using visual feedback of VLM as reward for reinforcement learning is not a novel method. [1][2]
2. The probability and other potential solution of reward hacking which is reported in the paper are not fully discussed.


[1] RL-VLM-F: Reinforcement Learning from Vision Language Foundation Model Feedback
[2] Learning Only with Images: Visual Reinforcement Learning with Reasoning, Rendering, and Visual Feedback

**Questions:**

1. What is the probability of reward hacking reported in the paper occurring during training? Is it an “always” result?
2. Intuitively, this reward hacking attack might be easily addressed by modifying the MLLM prompts that provide visual feedback, or by using methods based on visual embedding similarity to provide auxiliary judgment. Did the authors try this?

I will change my rating if the author resolves my confusion.

---

### Official Review · Reviewer_CYSt · 2025-11-01

**Soundness:** 2
**Presentation:** 2
**Contribution:** 1
**Rating:** 0
**Confidence:** 5

**Summary:**

This paper proposes an RL framework that incorporates for post training to LLMs using frozen VLMs. The authors propose reward signals, to guide this training. Through experiments the authors are able to show that the with finetuning Qwen3-8B the model is able to outperform other variants of Qwen3 all the way upto Qwen3-235B-A22B.

Below I enlist the strengths and weaknesses of this paper and and also do provide my rating in that context

**Strengths:**

#1: The method uses SOTA RL training strategies like GRPO to perform post training
#2: Result show that RL finetuning on Qwen3-8B outperforms a lot of open and closed source LLMs
#3: The paper uses strategies such as end of end VLM integration allowing for automation and scaling if needed in future

**Weaknesses:**

Major

#1: The method is inferior to the work presented in [1]. The work presented in [1] is also very similar. For the records: they do have expanded capabilities in addition to this work (Text-2-svg and Image-2-Svg) and GRPO based RLFT on top of SFT.

#2: The work presented in [1] also finetunes Qwen3-8B for text2svg tasks and they show robustness over multiple different datasets and previous works like [2,3] which makes this method inferior in the comparison.

#3: Also I would like to point out the related work which says [1] only relied on VLM-based image reconstruction and holistic visual similarity for rewards. This statement is factually wrong and I would encourage the authors strongly to read the paper and refine this part as in most of the rewards are non VLM based.

#4: In the same paper section 3.3 reward hacking is also discussed, as in this does not add in a lot of value according to me in this context

#5: I wonder why the authors do not use SFT followed by RL instead only use RL, [1] had already shown this issue and discussed the same.

#6: The comparison is too weak, I would have also included results RL with other architectures like Qwen 2.5, Llama, this gives me a hint that the paper is underprepared

#7: Details like human informed selection strategy and details related to those is not very obvious

#8: The dataset specially the test set seems a lot more less diverse and useful with only 164 samples. It would be intersting to see how this method works for different open world datasets as in [1,2]


Minor:

#1: Metrics are not clear

#2: Why is there only one kind of metric

#3: Training details like hyperparameter selection and other important details seems to be missing


Ref
1. Rodriguez, J. A., Zhang, H., Puri, A., Pramanik, R., Feizi, A., Wichmann, P., Mondal, A. K., Samsami, M. R., Awal, R., Taslakian, P., Gella, S., Rajeswar, S., Vazquez, D., Pal, C., & Pedersoli, M. (2025). Rendering-Aware Reinforcement Learning for Vector Graphics Generation. In NeurIPS 2025.

2. Jain, Ajay, Amber Xie, and Pieter Abbeel. "Vectorfusion: Text-to-svg by abstracting pixel-based diffusion models." CVPR. 2023.

3. Xing, X., Zhou, H., Wang, C., Zhang, J., Xu, D., & Yu, Q. (2024). Svgdreamer: Text guided svg generation with diffusion model. In CVPR 24.

**Questions:**

See Weaknesses

---

### Official Review · Reviewer_2EFg · 2025-11-01

**Soundness:** 1
**Presentation:** 2
**Contribution:** 2
**Rating:** 2
**Confidence:** 4

**Summary:**

This paper introduces a method to train LLMs to do the text to SVG task using reinforcement learning. Specifically the paper uses a vision-language model (VLM) as a visual reward model to see how well the _rendered_ SVG matches the user's text description. Using RL/GRPO introduced reward hacking, where the model learned to insert descriptive text (e.g., "Red Roof") into the image rather than rendering the actual graphics, as this still received a high score from the VLM. To mitigate this, the authors employ inference-time scaling, prompting the model to use an extended Chain-of-Thought "thinking mode" to decompose the visual task before generating code, which successfully prevented this hacking behavior.

**Strengths:**

- The paper's primary strength is its use of a vision-language model (VLM) as a "perceptual judge" to provide a reward signal, which is much better than pixel-wise + CLIP similarity style rewards used in pervious works.

- The authors identify reward hacking as a main issue with using VLMs as a judge and provide empirical evidence that using CoT can prevent reward hacking.

**Weaknesses:**

1. Equation (5) seems a little strange, why use this scheme when one could just inspect the logits of the VLM to get the full distribution. What if $\min R == \max R$ ?

1. The authors mentions in lines 96-98 and beyond that reward hacking was prevented by using chain-of-thought thinking. While you may have empirically shown that this may mitigate reward hacking, it doesn't make intuitive sense since a model trained with reinforcement learning with a faulty reward _will_ eventually learn to reward hack. The paper doesn't show much experimentation in finding out why using CoT prevents reward hacking.

1. Lines 460-462 mention, "[existing related work] is ill-suited for a task where a single prompt can have multiple valid visual outputs." Though, in Equation (4) and Line 239 the tuple $(q, I, I*)$, including the actual ground-truth reference rendered SVG is given to the VLM to judge. These two statements seem incompatible then.

1. While the data used to train the model may be technically _disjoint_ from the data used to evaluate the model, it is not clear why the improvements in the proposed method would not just come from having a very similar data distribution to the train set.

1. The baselines used were closed-source LLMs and open-sourced LLMs, there are no strong baselines of existing techniques specifically for text-to-SVG. For instance, a strong baseline could be a fine-tuned open-sourced LLM on the training data.

1. While the "Thinking" model successfully avoids this specific text-injection hack, the paper doesn't provide strong evidence that other, more subtle forms of reward hacking are not occurring.

**Questions:**

1. Regarding the VLM reward calculation in Equation (5), what was the reasoning for the "drop min/max and average" sampling strategy over querying the VLM's output logits directly? Additionally, how did this averaging scheme handle the edge case where all five sampled scores were identical (i.e., $\min R = \max R$)?

1. Were there any additional experiments done to explore why CoT prevents reward hacking?

1. In the related work discussion, you critique other methods as "ill-suited" for this task because a single prompt can have multiple valid visual outputs. However, your own reward function appears to be conditioned on a single "ground-truth" reference image $I*$. How does this approach avoid penalizing other valid SVG outputs that are semantically correct but visually different from the specific reference $I^*$?

---

### Note · Authors · 2026-01-05

I have read and agree with the venue's withdrawal policy on behalf of myself and my co-authors.